# Persistence of Metabolomic Changes in Patients during Post-COVID Phase: A Prospective, Observational Study

**DOI:** 10.3390/metabo12070641

**Published:** 2022-07-13

**Authors:** Peter Liptak, Eva Baranovicova, Robert Rosolanka, Katarina Simekova, Anna Bobcakova, Robert Vysehradsky, Martin Duricek, Zuzana Dankova, Andrea Kapinova, Dana Dvorska, Erika Halasova, Peter Banovcin

**Affiliations:** 1Clinic of Internal Medicine-Gastroenterology, University Hospital in Martin, Jessenius Faculty of Medicine in Martin (JFM CU), Comenius University in Bratislava, 036 01 Martin, Slovakia; liptak@jfmed.uniba.sk (P.L.); martin.duricek@gmail.com (M.D.); pbanovcin@gmail.com (P.B.); 2Biomedical Centre BioMed, Jessenius Faculty of Medicine in Martin (JFM CU), Comenius University in Bratislava, Mala Hora 4, 036 01 Martin, Slovakia; eva.baranovicova@uniba.sk (E.B.); zuzana.dankova@uniba.sk (Z.D.); andrea.kapinova@uniba.sk (A.K.); dana.dvorska@uniba.sk (D.D.); erika.halasova@uniba.sk (E.H.); 3Clinic of Infectology and Travel Medicine, University Hospital in Martin, Jessenius Faculty of Medicine in Martin (JFM CU), Comenius University in Bratislava, 036 01 Martin, Slovakia; simekova@jfmed.uniba.sk; 4Clinic of Pneumology and Phthisiology, University Hospital in Martin, Jessenius Faculty of Medicine in Martin (JFM CU), Comenius University in Bratislava, 036 01 Martin, Slovakia; bobcakova2@uniba.sk (A.B.); robert.vysehradsky@uniba.sk (R.V.)

**Keywords:** metabolome, post-COVID, SARS-CoV-2, COVID-19, NMR spectroscopy

## Abstract

Several relatively recently published studies have shown changes in plasma metabolites in various viral diseases such as Zika, Dengue, RSV or SARS-CoV-1. The aim of this study was to analyze the metabolome profile of patients during acute COVID-19 approximately one month after the acute infection and to compare these results with healthy (SARS-CoV-2-negative) controls. The metabolome analysis was performed by NMR spectroscopy from the peripheral blood of patients and controls. The blood samples were collected on 3 different occasions (at admission, during hospitalization and on control visit after discharge from the hospital). When comparing sample groups (based on the date of acquisition) to controls, there is an indicative shift in metabolomics features based on the time passed after the first sample was taken towards controls. Based on the random forest algorithm, there is a strong discriminatory predictive value between controls and different sample groups (AUC equals 1 for controls versus samples taken at admission, Mathew correlation coefficient equals 1). Significant metabolomic changes persist in patients more than a month after acute SARS-CoV-2 infection. The random forest algorithm shows very strong discrimination (almost ideal) when comparing metabolite levels of patients in two various stages of disease and during the recovery period compared to SARS-CoV-2-negative controls.

## 1. Introduction

In the last two decades, the metabolite profiling has been used to help understand pathophysiological processes in various diseases. Different biomolecules that form human metabolome may be affected by endogenous and exogenous factors, ultimately resulting in its dysregulation. Metabolic dysregulation is associated with a variety of diseases. Therefore, substantial research effort is focused on identifying promising biomarkers that could be useful for the diagnosis of different diseases or determination of their prognosis [1].

Recently published studies have shown changes in plasma metabolites in various viral diseases such as Zika, Dengue, RSV or SARS-CoV-1 [2,3,4]. Their occurrence in SARS-CoV-2 infection is therefore highly anticipated. Indeed, results of studies conducted so far are attributing importance to metabolomics status in COVID-19 patients in terms of early diagnosis, prognosis and prediction of the disease course [5,6]. Changes in the level of serum metabolites and lipidemic profile [7] in COVID-19 patients may reflect and reveal specific pathological processes that affect not only the lung tissue but entire organisms [8,9]. Despite some evidence that serum metabolite levels change during the acute SARS-CoV-2 infection, data describing their levels after the acute phase are scarce [10,11,12]. Therefore, the aim of this study was to analyze the metabolite profile of patients who suffered from a clinically moderate to serious course of COVID-19 approximately one month after the acute infection and to compare these results with healthy (SARS-CoV-2-negative) controls.

## 2. Results

Twenty-three metabolites and lipoprotein fraction were identified in denatured plasma in both patients and healthy subjects. Signals from 22 compounds were suitable for quantitative evaluation (Appendix A). To obtain a complex look at the metabolomics data, we employed PCA analysis that processes multidimensional data into 2D visualization and PLS-DA analysis that includes also a discriminatory algorithm. For both algorithms, the relative concentrations of metabolites in blood plasma determined by NMR were used as input variables. We avoided feeding the algorithms with intensities of 0.05–0.01 ppm binned NMR spectra as it is common in the metabolomic studies, since there may be regions of NMR spectra marked by the algorithms as important that are not unambiguously related to one single metabolite and therefore may be more hardly associated with the biological relevance.

When comparing all sample groups together, both algorithms showed an indicative shift in metabolomics features from sample group A (samples taken within 24 h of admission) over sample group B (5–8 days after admission) and sample group C (average 42 days after admission) towards the controls (Figure 1), where the time distance between the A-group and controls was the longest and group C partially overlapped with the control group. The PCA and PLS-DA analyses of metabolomics data comparing patients in the acute phase of COVID-19 and controls was the subject of previous studies [5]. In this study, we focused on the binary comparison of the systems in post-COVID-19 patients (sampling C) against controls. Here, both algorithms showed that the separation of the groups is attainable, but not ideal (Figure 2). The calculated VIP scores from PLS-DA analysis for both showed that systems are included in Appendix A.

The null hypothesis of equality of population medians among groups was tested by the non-parametrical Kruskal–Wallis test for multiple comparison with Dun’s post hoc test for pairwise comparison. Both tests are designed for non-normal distribution, which is assumed since the normality cannot be reliably tested in groups of size under 30. The results from statistical evaluation of metabolomics data are shown in Table 1, the corrected α using Bonferroni’s correction method was 0.008333 (threshold for *p*-value to be considered significant). Relative percentage changes were derived from medians. No significant changes among groups were observed for plasma lactate and lysine level.

One of the aims of the study was the evaluation of the classification power of the binary systems patients-controls. For this, we employed cross-validated random forest (RF) algorithm to obtain a more realistic estimation of the discriminatory power of the systems, since RF algorithm is known to be robust to outliers [13] and it does not tend to overfit the data in comparison to PLS-DA [13,14]. As input variables, relative concentrations of metabolites in blood plasma determined by NMR spectroscopy were used (lipoprotein fraction was omitted because of not meeting the criteria of metabolite). Results are summarized in Table 2, where we rated the performance after the area under curve (AUC) derived from the receiver operator characteristic curve (ROC), as recommended by Xia et al. [15]. Besides that, we used the Matthews correlation coefficient (MCC), calculated from the confusion matrix, that is, intuitive and straightforward: to get a high-quality score, the classifier has to make correct predictions both on the majority of the negative cases, and on the majority of the positive cases, independently of their ratios in the overall dataset [16].

With the increasing number of variables included in the algorithm, the discriminatory performance increased, as expected. Based on AUC values, the ideal discrimination with AUC equal to 1 was achieved for the system A-controls. Binary systems B-controls and C-controls were discriminated almost ideally with an AUC of 0.99, when evaluated by AUC value.

Matthew correlation coefficients revealed that the ideal discrimination is attainable only for the system A-controls, where MCC = 1. For systems B-controls and C-controls, MCC = 0.936 or 0.895, respectively revealed weaker, but still promising discrimination. After repeated RF runs, metabolites permuted in the importance order. Histidine appeared in all analyses as a metabolite with a very high discriminatory power. The subsequent analyses, where histidine was omitted, also achieved very high AUC (A, B, C vs. controls: 0.99, 0.94 and 0.96), implying that histidine plays a supporting but not key role for the RF performance. Very similarly, after omitting both, histidine and proline, high AUCs were achieved (A, B, C vs. controls: 0.98, 0.95, and 0.93). In other words, the discriminatory power of the system does not rely on one or two sole metabolites.

## 3. Discussion

One of the most known and prevalent factors influencing COVID-19 disease progression [17] is hyperglycemia. COVID-19 disease may cause prolonged uncontrolled elevated blood glucose level [18] due to COVID-19-induced insulin resistance [19] and COVID-19-impaired insulin production [20]. Contrary to COVID-19 non-survivors, patients with positive COVID-19 outcome showed the ability to normalize energy metabolism during the first week of hospitalization. [5]. The hyperglycemic condition in the first days of hospitalization could be, besides others, attributed to the treatment with dexamethasone. This synthetic glucocorticoid with anti-inflammatory properties impairs glucose metabolism via stimulation of hepatic gluconeogenesis from amino acids released from muscles and inhibition of glucose uptake [21]. It was observed that even a single dose of 10 mg dexamethasone could temporarily increase blood glucose level [22]. After the reduction of steroid doses, drug-induced diabetes is expected to resolve [23]. As expected, patients included in our study showed significantly increased plasma glucose level on the first day of sampling that decreased over time, but remained increased up to circa 20% against controls one month after hospital discharge. This indicates persistent alterations in the glucose metabolism. Decreased glucose utilization over the whole monitoring time was also supported by decreased plasma levels of glycolytic product pyruvate in COVID-19 patients (Figure 3). Increasing evidence suggests that pyruvate plays a crucial role as an immunonutritional factor for polymorphonuclears (PMN) and other white blood cells (WBC), e.g., granulocytes that are largely involved in the immune response regulation. According to the available clinical data and experience, the course of infection most often worsened within 7 days of the patient’s admission and was related to gradual development of inflammatory changes in lung tissue. We believe that the observed odd trend is related to the collection of sample B at the peak of the disease, when patients mobilize all reserves and cells of the immune system, with high demand for their nutrition and survival [24].

In times of impaired glycolysis, other energy substrates are expected to maintain body requirements. The switch to ketone bodies metabolism in time of insufficient glucose utilization is often observed in various stress conditions, as it was observed previously in COVID-19 patients during hospitalization [5]. The initial increase in ketone body 3-hydroxybutyrate on the first day suggested a ketotic-like state that was reduced over day 7, but still persisted to a small extent at a follow-up examination. Ketone bodies accumulation in COVID-19 patients may be related to a reduced hepatic capacity to oxidize acetyl-CoA in the mitochondria, which is then redirected to the synthesis of acetoacetic acid and 2-hydroxybutyric acid. In parallel with that, a plasma level of lipoprotein fraction consisting of more than 30% triacylglycerols showed a reverse time course to 3-hydroxybutyrate (Figure 3), as these are in a mutual relation of substrate-product, since the ketone bodies are produced predominantly from fatty acid oxidation-derived acetyl-CoA. Besides basal function as an energy source for the brain, heart and skeletal muscle, 3-hydroxybutyrate plays a key role as a signaling mediator [25], driver of protein post-translational modification [26], and modulator of inflammation, showing predominantly anti-inflammatory, but also pro-inflammatory, effects [27,28]. These and many other functions of 3-hydroxybutyrate make it an important agent in the acute as well as recovery phase in patients. We observed increased levels of succinate in samples A and C and no percentual change in sample B. Some hypotheses suggest that succinate has potent antiviral activity [25], thus the reduction in the level of succinate in sample B in our study may be explained by this activity during continuous severe infection.

In the post-COVID-19 phase, we observed the tendency of the plasma levels of several further metabolites to normalize towards the levels of controls. Glutamine serves, besides others, as a fuel for immune cells: lymphocytes, neutrophils, and macrophages [29] and plays a crucial role in the production of cytokines [30]. In an acute inflammatory condition, the demand for glutamine increases, and if the endogenous synthesis of glutamine does not fulfil the requirements of the body, this may lead to a decrease in its plasma levels. In our study, the plasma levels of glutamine restored relatively quickly after an initial decrease and remained unchanged in the post-COVID phase (Figure 3). In a previous study by Cengiz et al., the administration of glutamine in the early period of infection suggested a shortened hospital stay due to COVID-19 and decreased the need for ICU stay [31]. The patients included in our study were able to balance the level of glutamine in the blood relatively early, which could have contributed to their recovery.

The temporal changes of plasma levels of BCAAs including leucine, isoleucine and valine, together with their corresponding ketoacids (BCKAs), showed very similar pattern, namely, increase in the first hospitalization days that normalized towards the control levels during one month after hospital discharge (Figure 3, boxplot shown for leucine and ketoleucine). Their common biochemical pathways are manifested on the heatmap, where strong relations between BCAAs and BCKAs are shown (Appendix A). BCAAs are closely connected to glucose metabolism, as they are (not causative) markers of a loss of insulin action [26], which supports the above-discussed findings. The observed significant increase in blood plasma levels of leucine, isoleucine and valine in the acute COVID-19 phase (A and B sampling) may indicate their restricted degradation that is controlled by a number of tissues, particularly by the muscles and liver. The BCAAs status also convey the ability of the organism to maintain the vital functions of protein synthesis and breakdown and BCAAs accumulation may signalize their limited utilization with the negative effect on protein expression. Besides that, there is an established association between the elevated circulating BCAAs and their deleterious effects, as their increased concentration may promote oxidative stress and inflammation [27], having also a neurological impact [28].

COVID-19 patients showed decreased histidine plasma levels in comparison to controls over the whole evaluation time. L-histidine, which cannot be synthetized de novo in humans and which is sourced mainly from the diet, exhibits antioxidant capabilities [32], and it is able to suppress the accumulation of IL-6 and TNF-α mRNA in a dose-dependent manner and could directly affect the regulation of pro-inflammatory cytokines [33]. In COVID-19 infection, cytokine storm is associated with the development of severe pneumonia requiring oxygen supplementation, which can finally lead to respiratory failure of the patient. Cytokine storm correlates mostly with the levels of pro-inflammatory cytokines, which are increased. This can explain such an odd trend in sample B, which was collected at the peak of the infection [34]. However, the immune response is known to be controlled also by histamine, an amine produced exclusively by histidine decarboxylation. The pleiotropic actions of histamine due to the different natures of its receptors allow to exert broad and oppose effects on the immune system, histamine balances important inflammatory reactions as well as immunomodulation [29]. Given the fact that both histidine and its main metabolic product histamine participate in immune processes, the exact biochemical mechanisms could be worthy of a deeper analysis and examination in the complex immune response.

Plasma levels of alanine were decreased only on the first sampling day, and later they increased towards controls. Alanine, together with BCAAs and glutamine, are a major part of inter-organ nitrogen shuttle. It could be suggested that, simultaneously with other processes, an insufficient utilization of BCAAs may lead to a reduced availability of amino groups needed for the synthesis of alanine and glutamine. Besides this, we observed a slightly increased creatinine level in COVID-19 patients in all sampling times, which may be linked to dehydration due to strong inflammation; however, impaired renal function cannot be excluded. Interestingly, this trend did not change over time and the patients in the post-acute phase showed a remaining increased level of blood creatinine. The metabolomics co-partner, creatine, was significantly increased only in one sampling time after one week of hospitalization. Patients forced to lie in bed for a substantial time period lacked spontaneous movement utilizing muscle energy, which is probably the reason for the increase of plasma creatine solely at this time point.

Perturbations in phenylalanine and tyrosine biosynthesis were recognized in SARS-CoV-2 positive patients previously by Barberis et al. [30]. In our study, we observed increased phenylalanine levels in COVID-19 patients during the hospital stay, but normalized levels in the later post-COVID phase. Besides being a proteinogenic amino acid, phenylalanine is irreversible hydroxylated to tyrosine in kidneys and liver and tyrosine is further used for catecholamine synthesis. In our work, we did not observe a tyrosine decrease in COVID-19 patients that would indicate primary impaired hydroxylation from phenylalanine, suggesting that the utilization of phenylalanine is restricted by other regulation mechanisms.

In summary, the metabolomics changes observed on the first day of sampling gradually subsided, where some metabolites reached the level of control, but some remained altered even 42 days (mean) after the first sampling. A limitation of this study could be a relatively small cohort of patients, although we tried to correct this by choosing a homogenous population. Another limitation is a rather long-range interval of sample C collection. This is due to different durations of hospitalization of the patients. From a clinical point of view, all patients on a clinical medical check-up during the collection of sample C presented the same spectrum of post-COVID recovery symptoms (slightly higher exhaustion after basic physical activity than before COVID, dyspnea after more intensive physical activity comparably worse than before COVID, no requirement of oxygen supplementation, no gastrointestinal or neurological symptoms). The homogeneity of the group is also further supported by standard biochemical and haematological examination, which does not show any relevant or statistically significant differences between patients.

### Multivariate and Discriminatory Analyses

PCA as well as PLS-DA analyses (Figure 1 and Figure 2) both showed obvious differences between metabolomic data from patients in the COVID-acute phase (group A and B) and controls. Interestingly, a relative shift of metabolomics data from Group A over Group B and C towards controls can be well traced, visualizing the metabolomic recovery over time. The variables with the highest VIP score in PLS-DA were: lipoproteins, glucose, alanine and glutamine in A-B-C-controls comparison and lipoproteins, glutamine, alanine and leucine in the C-controls comparison. Given that the lipoprotein fraction, that scored as highest, cannot be considered as a metabolite, we omitted it in the next analyses when searching for the possibilities of correct classification.

The high potential of metabolomics in the field of biomarkers was already demonstrated by successful discrimination of COVID-19 patients in the acute phase against controls [5]. In this work, the employed RF algorithm used included cross-validation via balanced subsampling. It worked with two-thirds of the data for training and the rest for testing for regression, and about 70% of the data for training and the rest for testing during classification to overcome the negative aspects of training and testing on the same data. This approach partially substitutes the validation on an independent dataset; however, it cannot fully replace the clinical validation. We used relative concentrations of metabolites in plasma expressed by the spectral integrals of particular NMR regions as input variables for RF algorithm. In the case of highly correlating predictors, RF may label some of them as unimportant, so RF was ran ten times. Within the RF re-runs, metabolites permuted in the importance order a little.

In our study, massive metabolomics changes in acute COVID-19 resulted in the ideal discrimination for binary system Group A-controls (AUC = 1, MCC = 1) and slightly weaker, but still very promising discrimination, for the system B-controls (AUC = 0.992, MCC = 0.936) (Table 2, Figure 4). The post-COVID metabolomics changes and related discriminations are less known. In this study, we show that the persisting metabolomics alterations in the post-COVID phase are strong enough to discriminate post-COVID patients from controls with very high specificity and sensitivity; in our patients group achieving AUC = 0.991. Although the AUC parameter indicates that the lastly mentioned binary system can be separated almost ideally, it obtained an out-of-bag error of 3/62 results to an error of correct classification of 5%. The further parameter, the proportion of the correct predictions among the total number of cases examined, achieved averaged accuracy, which for this system equaled 0.932. The last parameter expressing the classification performance was MCC (a coefficient of 1 would represent perfect, and of 0 random prediction), which showed an encouraging value of 0.895, but not ideal results.

In some patients, post-acute sequelae of SARS-CoV-2 infection may persist for a long time, with significant consequences for their further quality of life. Understanding the pathophysiological processes at the enzymatic level, which individually take place not only in the acute phase of infectious diseases, but also in the stage of convalescence, appears to be an important milestone for the future. Metabolomics may contribute to a better understanding of these processes in the future, which may ultimately benefit the patient.

## 4. Materials and Methods

The study was performed from November 2021 until February 2022. Twenty-five (25) patients were enrolled in the study. All participants signed the informed consent. The study was approved by the Ethical Committee of the Jessenius Faculty of Medicine in Martin, Comenius University, Slovakia (Certification code at the US Office for Human Research Protection, US Department of Health and Human Services: IRB00005636 Jessenius Faculty of Medicine, Comenius University in Martin IRB # 1) with identification number: EK 65/2021. Every patient enrolled in the study had SARS-CoV-2 infection confirmed by a polymerase chain reaction (PCR). The study focused on patients with typical SARS-CoV-2 infection symptoms (fever, cough and dyspnea) in order to achieve a high level of homogeneity (in the means of the COVID-19 presentation) within the cohort. Therefore, only hospitalized patients with a severe course of COVID-19, requiring oxygen supplementation but not invasive artificial pulmonary ventilation [31], (based on the National Institutes of Health/NIH/criteria) with X-ray or CT confirmed pneumonia were considered for the study. The characterization of the patient group is shown in Table 3. Only patients with compensated chronical diseases were considered for the study. Exclusion criteria were age under 18 years, pregnancy and unwillingness or incapability to sign the informed consent. Blood samples from age-matched 37 subjectively healthy volunteers from our internal biobank, age 51 ± 16 yrs (female/male = 12/25), sampled before the COVID pandemic (in the years 2018–2019), were used as controls.

Peripheral blood was used for the analysis. Blood samples were collected at aseptic conditions on 3 occasions. Each time, two 10 mL EDTA (Ethylenediamin tetra-acetic acid) collection tubes were used. The first sample (Sample A) was taken within 24 h of admission to the hospital. The second sample (Sample B) was taken on days 5–8 (based on the course of the hospitalization, e.g., early discharge would lead to a shorter interval between Samples A and B) and the third sample (Sample C) was taken 42 days (29–54 days) after the first sample. Blood for standard biochemical and hematological analysis was obtained during every sample-taking occasion (A, B, C), Table 4.

The groups of samples were labeled Group A, Group B and Group C based on the sample labeling (A, B or C) for further analysis.

### 4.1. Sample Preparation

Blood was collected in EDTA-coated tubes, centrifuged at 4 °C, 2000 rpm (380× *g*-force), for 20 min. Plasma was deproteinized according to Gowda et al. [35]. The mixture obtained after adding 600 μL of methanol to 300 μL of plasma was shortly vortexed and stored at −20 °C for 20 min. After centrifugation at 14,000 rpm (14,800× *g*-force), 700 μL of supernatant were dried out. Before measurement, the dried matter was carefully mixed with 100 μL of stock solution and 500 μL of deuterated water. Then, 550 μL of final mixture were transferred into 5 mm NMR tube. Stock solution consisted of: 100 mM phosphate buffer (pH-meter reading 7.40) and 0.30 mM TSP-d_4_ (3-(trimethylsilyl)-propionic-2,2,3,3-d_4_ acid sodium salt) as a chemical shift reference in deuterated water.

### 4.2. NMR Data Acquisition

NMR (Nuclear Magnetic Resonance) data were acquired on a 600 MHz NMR spectrometer Avance III from Bruker equipped with TCI (triple resonance) cryoprobe. Initial settings were done on an independent sample and adopted for measurements. Samples were stored in Sample Jet at approximately 6 °C before measurement for a maximum of 2 h. We used standard Bruker profiling protocols that we modified as follows: profiling 1D NOESY with presaturation (noesygppr1d): FID size 64k, dummy scans 4, number of scans 128, spectral width 20.4750 ppm; COSY with presaturation was acquired for randomly chosen 20 samples (cosygpprqf): FID size 4k, dummy scans 8, number of scans 1, spectral width 16.0125 ppm; homonuclear *J*-resolved (jresgpprqf): FID size 8k, dummy scans 16, number of scans 4; profiling CPMG with presaturation (cpmgpr1d, L4 = 126, d20 = 3 ms): FID size 64k, dummy scans 4, number of scans 128, spectral width 20.0156 ppm. All experiments were conducted with a relaxation delay of 4 s; all data were once zero-filled. An exponential noise filter was used to introduce 0.3 Hz line broadening before Fourier transform. Samples were measured at 310 K and randomly ordered for acquisition.

### 4.3. Data Analysis

Spectra were solved using internal metabolite database, online human metabolome database (www.hmdb.ca) [36], chenomx software free trial version and literature [35]. For all compounds, the multiplicity of peaks was confirmed in J-resolved spectra, homonuclear cross peaks were confirmed in cosy spectra (Appendix A). A chemical shift of 0.000 ppm was assigned to TSP-d_4_ signal. All spectra were binned to bins of the size of 0.001 ppm, starting from 0.500 ppm to 9.500 ppm, with excluded water region 4.6–4.9 ppm and EDTA region 3.07–3.62 and 3.6–3.62 ppm. No normalization method was applied on NMR data, as we took exactly the same amount of blood plasma from all samples. In 0.001 ppm binned spectra, we summed bin intensities expressing the integrals of signals in the spectra subregions without overlapping peaks, i.e., assigned to sole particular metabolites. These values were used as a relative concentration of metabolites in blood plasma. Metabolites not having appropriate signals for the quantitative evaluation, or if their peak assignment was not unambiguous, were excluded from further processing. The NMR spectra of patients in various sampling times and controls are shown in Appendix A.

The null hypothesis of equality of population medians among groups was tested by the non-parametric Kruskal–Wallis test with Dun’s post hoc test for pairwise comparison. Principal component analysis (PCA) and the receiver operating characteristic curves (ROC) derived from random forest (RF) algorithm were performed using Metaboanalyst [37].

Note: In this work, we use common labelling BCAAs for branched-chain amino acids: leucine, isoleucine and valine, and BCKAs for their 2-oxoderivates, branched-chain keto acids: ketoleucine (2-oxoisocaproate), ketosioleucine (3-methyl-2-oxopentanonate) and ketovaline (2-oxoisovalerate), as well as mentioned trivial names of BCKAs that better evoke their origin.

## 5. Conclusions

Significant metabolomic changes persist in patients more than a month after acute SARS-CoV-2 infection based on the presented data. This is predominantly defined by increased plasma levels of glucose and 3-hydroxybutyrate and decreased plasma level of acetate and histidine with normalization of BCAAs and BCKAs. Random forest algorithm is showing very strong discrimination (almost ideal) obtained when comparing metabolite levels of patients in 2 various stages of disease and during the recovery period compared to SARS-CoV-2-negative controls. It is therefore possible to conclude, that, based on our data, patients with moderate-to-serious course of COVID-19 approximately one month after acute infection are not fully recovered and it is possible to identify these patients based on metabolomic profile with 95% specificity and 95% sensitivity when compared to healthy (SARS-CoV-2-negative controls).

## Figures and Tables

**Figure 1 metabolites-12-00641-f001:**
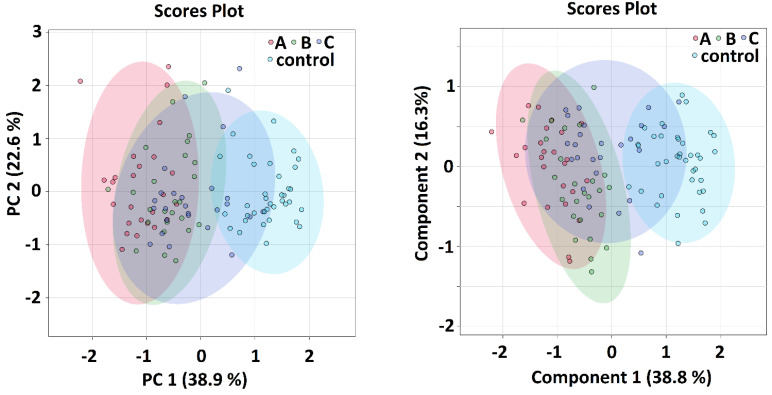
PCA (**left**) and PLS–DA (**right**) analysis of patients with COVID–19 disease at three various sampling times in patients and controls, the relative concentrations of metabolites in blood plasma were used as input variables.

**Figure 2 metabolites-12-00641-f002:**
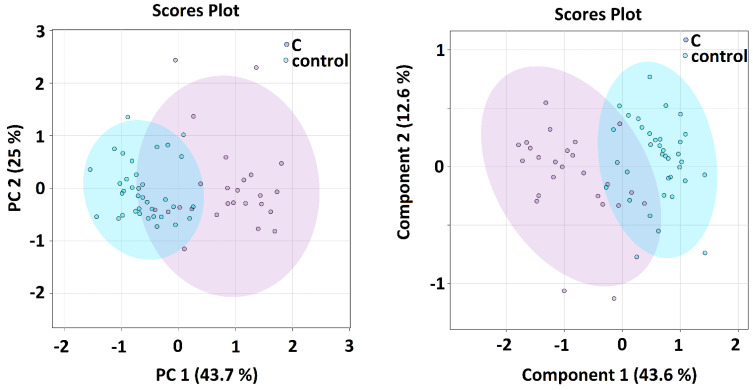
PCA (**left**) and PLS–DA (**right**) analysis of patients with COVID–19 disease at sampling time C in average 42 days after hospitalization in patients and controls, the relative concentrations of metabolites in blood plasma were used as input variables.

**Figure 3 metabolites-12-00641-f003:**
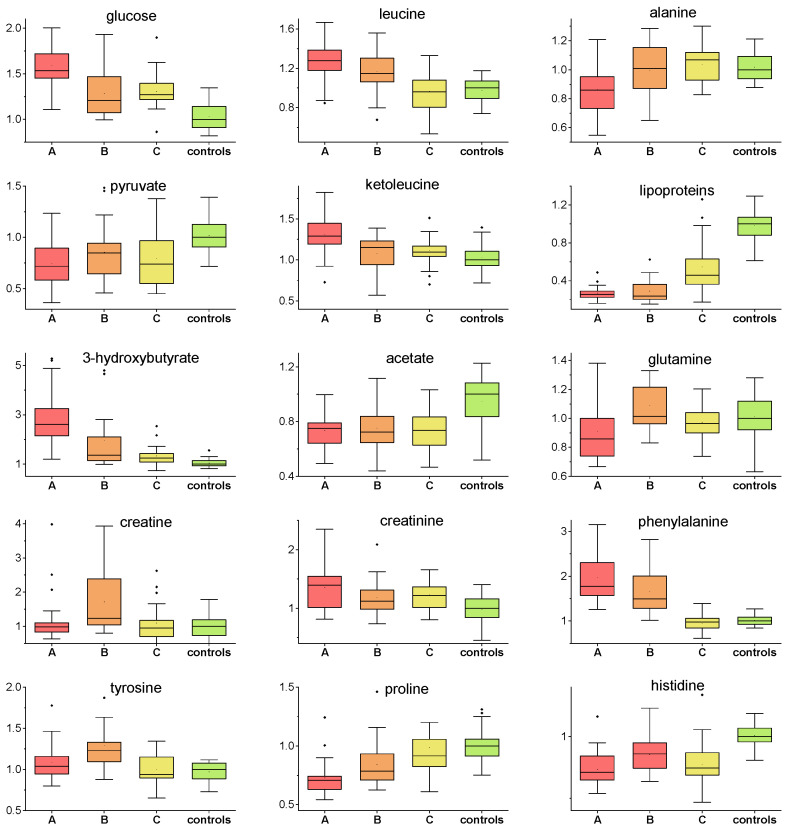
Relative concentrations of plasma metabolites in patients in three various sampling times and controls, values related to median of controls set to 1.

**Figure 4 metabolites-12-00641-f004:**
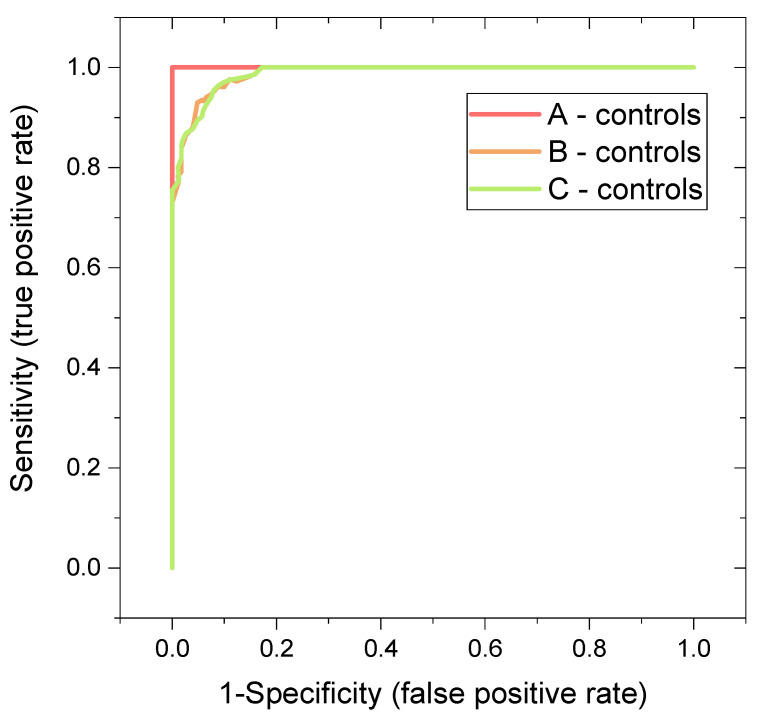
ROC curves derived from random forest discriminatory analysis for binary systems: patients in various sampling times versus controls: A-First day of hospital admission, B-In average 7 days after hospital admission. C-In average 42 days after hospital admission, relative concentrations of plasma metabolites were used as input variables. All evaluated metabolites were used as input variables; more details in Table 2.

**Table 1 metabolites-12-00641-t001:** Statistical evaluation based on the Kruskal–Wallis test for multiple comparison and post hoc Dun’s tests for pairwise comparison, percentual change derived from medians, algorithms used relative concentrations of metabolites in blood plasma.

	Kruskal–Wallis	A-Controls		B-Controls		C-Controls	
	*p* Value	*p* Value	% Change	*p* Value	% Change	*p* Value	% Change
alanine	0.0071	0.0031	−15	0.85	x	0.71	x
valine	0.00082	0.00099	15	0.00051	16	0.012	x
glucose	0.00011	6.3 × 10^−6^	54	0.20	21	0.15	26
leucine	8.4 × 10^−7^	9.6 × 10^−7^	37	0.00077	15	0.63	x
isoleucine	2.6 × 10^−7^	2.6 × 10^−8^	32	0.00029	21	0.0016	17
acetate	0.0000092	0.00037	−25	2.5 × 10^−5^	−28	5.2 × 10^−5^	−28
pyruvate	0.000020	4.4 × 10^−6^	−28	0.0066	−16	0.00049	−26
citrate	6.4 × 10^−10^	0.041	x	0.00030	−26	0.00078	21
phenylalanine	1.1 × 10^−14^	2.4 × 10^−9^	77	7.6 × 10^−8^	49	0.40	x
tyrosine	0.000051	0.042	x	6.1 × 10^−6^	22	0.52	x
glutamine	0.0089	0.024	−15	0.24	x	0.17	x
lipoproteins	1.1 × 10^−16^	1.4 × 10^−13^	−75	1.9 × 10^−13^	−77	0.00014	−54
ketoleucine	0.00036	3.0 × 10^−5^	29	0.37	x	0.060	x
ketoisoleucine	0.00025	6.5 × 10^−5^	37	0.059	x	0.0011	19
ketovaline	0.043	0.0057	20	0.11	x	0.090	x
3-hydroxy-butyrate	1.6 × 10^−12^	9.3 × 10^−14^	261	8.4 × 10^−6^	34	0.0010	24
creatine	0.0028	0.82	x	0.00067	25	0.81	x
creatinine	0.0025	0.0040	39	0.052	x	0.00066	22
histidine	4.0 × 10^−10^	5.8 × 10^−10^	−29	0.00026	−15	2.8 × 10^−7^	−26
succinate	0.0013	0.0083	28	0.023	x	0.00020	48
proline	3.3 × 10^−8^	8.6 × 10^−9^	−30	0.00018	−22	0.14	x

**Table 2 metabolites-12-00641-t002:** Result from RF discriminatory analyses of binary systems patients-controls, AUC values derived from ROC curve, MCC–Matthews correlation coefficient.

System	Features	Oob Error (Based on Predicted Class Probabilities)	Average Accuracy Based on 100 Cross-Validations	AUC	MCC
A-control	3 most important metabolites: histidine, proline, 3-hydroxybutyrate	0	0.999	1	1
	5 most important metabolites: histidine, proline, 3-hydroxyburtyrate, acetate, citrate	0	0.999	1	1
	all evaluated metabolites	0	1	1	1
B-control	3 most important metabolites: histidine, proline, 3-hydroxybutyrate	5/62	0.884	0.966	0.839
	5 most important metabolites: histidine, proline, 3-hydroxyburtyrate, pyruvate, citrate	5/62	0.923	0.981	0.839
	all evaluated metabolites	2/62	0.948	0.992	0.936
C-control	3 most important metabolites: histidine, glucose, pyruvate	5/62	0.929	0.969	0.829
	5 most important metabolites: histidine, glucose, pyruvate, phenylalanine, glutamine	5/62	0.929	0.987	0.829
	all evaluated metabolites	3/62	0.932	0.991	0.895

**Table 3 metabolites-12-00641-t003:** Characteristics of patients enrolled in the study.

	Median (IQR)
	Patients n = 25
Age [years]	58 (21)
Sex: Female/Male	7/18
Weight [kg]	82.6 (26)
Height [cm]	171 (8)
BMI	29 (9)
Chronic liver disease	3
Chronic kidney disease	3
Ischemic cardiac disease	3
Diabetes Mellitus	3
Thyroidal disease	4
Rheumatic disease	0
Other relevant	NA

**Table 4 metabolites-12-00641-t004:** Standard biochemical and hematological results of patients at the sampling times, median (IQR).

	Samples A	Samples B	Samples C	*p* Value (Multiple Comparison)
Na	133.32 (5.5)	140 (6)	139.4 (3.0)	<0.001
K	3.972 (0.6)	4.2 (0.65)	4.2 (0.45)	0.017
Cl	99.48 (6.0)	104 (6)	104 (3.0)	<0.001
Glucose	8.068 (1.35)	5.8 (3.05)	5.6 (1.5)	0.0021
Cretinine	83 (38.5)	60 (22.5) one missing	68 (32.5)	0.32
CRP	116.5 (123.4)	16.6 (31.45) one missing	2.2 (4.55)	<0.001
AST	1.2604 (0.68)	0.92 (1.13) eight missing	0.508 (0.285) one missing	<0.001
ALT	1.0365 (0.715) one missing	1.465 (1.325) nine missing	0.575 (0.49) one missing	<0.01
GMT	1.6815 (1.715) five missing	1.47 (2.48) three missing	0.815 (1.08) one missing	0.037
Bilirubin	10.7 (5.85)	11.4 (6.65) eight missing	9.6 (6.3) one missing	0.41
Leukocytes	6.7 (3.05)	7.7 (3.75) one missing	8.2 (2.5)	0.029
Hemoglobine [g/l]	142 (13)	135.5 (18.5) one missing	138 (13.5)	0.16
Platelets count	190 (162.5)	360.5 (215.5) one missing	259 (133)	<0.01

## Data Availability

All data including raw and evaluated NMR spectra are available on request: eva.baranovicova@uniba.sk.

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
