# Peer review of "Persistence of Metabolomic Changes in Patients during Post-COVID Phase: A Prospective, Observational Study"

_metabolites, 2022, doi:10.3390/metabo12070641_

Round 1
Reviewer 1 Report
The manuscript titled “Persistence of metabolomic changes in patients during post-2 COVID phase: a prospective, observational study” and submitted by Rosolanka is suitable for publication in Metabolites only after an exhaustive revision. In my opinion, it is necessary a more rigorous investigational approach even if the conceptualization is quite correct.
Some suggestions and lacks
-Recruited patients were divided depending on disease severities, please insert symptoms.
-The manuscript lacks of a PCA and PLS-DA analysis on patients with COVID differentiated with respect to the disease severities.
-Concerning the NMR experiments authors must consider that for a quantitative analysis through NMR spectroscopy (qNMR) a relaxation delay of 4 s is not appropriate. In details, for quantitative analysis using TSP as internal standard many researchers suggested to set D1 at 30 – 40 s for a correct analysis.
-Moreover, you wrote that TSP was used as a chemical shift reference in deuterated water without specifying with respect to those you have done the quantification of metabolites.
Remember that the internal TSP signal enabled the determination of absolute concentrations for identified metabolites.
-You wrote: we chose the spectra subregions with only single metabolite. Please, you have to explain what means….probably you are talking about not overlapping signals?
- No NMR spectra are reported. It is necessary add some data as an example.
- the numbering of table 3 and 4 is not correct in the text.
- Finally, consider that ketone bodies are produced predominantly in the liver from fatty acid oxidation-derived acetyl-CoA, with consequence on the serum accumulation in COVID-19 patients, probably due to a reduced hepatic capacity to oxidize acetyl-CoA in the mitochondria, which is then redirected to the synthesis of acetoacetic acid and 2-hydroxybutyric acid.
Reviewer 2 Report
Liptak et al. report on a NMR-based metabolome analysis of blood plasma of 25 patients with moderate to serious course of COVID-19. Samples were collected at admission, during hospitalization (5-8 days) and approximately one month after the acute infection (29-54 days), and compared with healthy controls. The use of the Random Forest algorithm gave a fairly good discrimination between controls and different sample groups, revealing that patients who suffered from moderate to serious course of the COVID-19 are not fully recovered one month after acute infection.
Despite the clear relevance of the topic, I feel the scientific protocol is not scientific sound or at least not exhaustively justified, and some further discussion is needed. More in detail, to deserve publication in Metabolites, the authors should address the following points:
Major points:
- 1. I am confused on the identified metabolites. Authors write that 24 metabolites were identified and 22 compounds were suitable for quantitative evaluation (lines 57-59), and these are actually listed in Table S1. However, only 21 entries are reported in Table 1 and only 15 boxplots are shown in Fig. 3. Is there a reason for that? Also, I believe that sorting the lists of metabolites in Table 1 and S1 based on a given order – alphabetically for instance - would be beneficial for the reader.
- 2. The interval for sample collection of group C is quite large (29-54 days). Is this the reason of the broadness observed in the score plot for group C? How different would the results be if narrower ranges are considered? I urge the authors to further discuss this point.
- 3. Among the characteristics reported in Table 3 for the 25 patients enrolled in the study there are a few diseases. Did any correlation with the change in the metabolomic profile caused by COVID-19 emerge? Were comorbidities considered in data evaluation?
- 4. I have some reservations on the protocol followed for the NMR data analysis. Authors claim they “avoided feeding the algorithms with binned NMR spectra as it is common in the metabolomic studies, since there may be regions of NMR spectra marked as important that are not straightforward and unambiguously related to biological relevance” (Lines 62-65). I partially disagree, as this is exactly the aim of an untargeted analysis, i.e. identifying what cannot be easily identified by a targeted analysis, and explore further the most relevant regions. As the authors rather identified known metabolites in the NMR spectra, I do not understand why they used subregions of the binned spectra and relative concentrations in the statistical analysis. I would consider in that case the possibility to exactly quantify the metabolites found in the blood plasma, using an internal reference with known concentration (TSP-d4 for instance). Authors should prove why feeding the algorithms with binned NMR would give unreliable/useless results, i.e. outputs should be provided, to give real proof of the suitability of their procedure.
- 5. A quite detailed discussion on the changes in metabolomic profile is given in the discussion. I find, however, that not enough emphasis has been given to some (to me) odd trends:
o pyruvate and histidine: decrease in sample A and C wrt to controls more marked than in sample B;
o ketoisoleucine, creatinine and (to a lesser extent) succinate: increase in sampes A and C wrt to controls, while no %change in samples B.
- 6. A comparative discussion with already published studies on the metabolic profiles of COVID-19 disease would improve the manuscript. For instance ref 8 reports the plasma metabolomic profile three months after discharge, how (dis)similar are the results with those observed here?
Minor points:
- 7. I recommend to add representative NMR spectra with metabolite assignment as supplementary material.
- 8. If a fine multiplicity (d, t, q, dd) is identified in Table S1, I do not see why a J-coupling should not be measured. The authors should complete Table S1.
- 9. The resolution of Figures 1 and 2 is quite poor and text is unreadable.
- 10. The authors should pay attention to some sentence structures, i.e. lines 132-133 and line 143.
- 11. The authors should pay attention to Table numbering, i.e. Table 1 is rather Table 3 (lines 322-323); Table 3 should be Table 4 (line 334).
Reviewer 3 Report
Persistence of metabolomic changes in patients during post- COVID phase: a prospective, observational study
This work show significant metabolomic changes persist in patients more than a month after acute SARS-CoV-2 infection. Random forest algorithm shows very strong discrimination (almost ideal) when comparing metabolite levels of patients in two various stages of disease and during recovery period compared to SARS-CoV-2 negative controls.
Introduction
It is necessary to talk about the possible metabolomic changes of covid19 on positive patients.
See references, for example:
Wu, D., Shu, T., Yang, X., Song, J. X., Zhang, M., Yao, C., ... & Zhou, X. (2020). Plasma metabolomic and lipidomic alterations associated with COVID-19. National Science Review, 7(7), 1157-1168.
Jia, H., Liu, C., Li, D., Huang, Q., Liu, D., Zhang, Y., ... & Liang, H. (2022). Metabolomic analyses reveal new stage-specific features of COVID-19. European Respiratory Journal, 59(2).
Xu, J., Zhou, M., Luo, P., Yin, Z., Wang, S., Liao, T., ... & Jin, Y. (2021). Plasma metabolomic profiling of patients recovered from coronavirus disease 2019 (COVID-19) with pulmonary sequelae 3 months after discharge. Clinical Infectious Diseases, 73(12), 2228-2239.
2. Results and discussion
Figure 1 and 2 are of low quality.
It is necessary to use the VIP score, to observe the signals that correspond to the differential metabolites between the different times and the control.
It is necessary to include a comparison of the nmr spectra of the controls at the different times.
In supplementary material introduce the figures of the different NMR spectra.
A heat map and a dendrogram would support the discussion of the work.
Conclusions
Not comment.
References
Not comment

Round 2
Reviewer 1 Report
The manuscript was improved enough, however an extensive editing of English language is required as the style is not correct or the sentences are not well written (e.g. ros 61-63 “For both algorithms, , relative concentrations of metabolites in blood plasma were used determined by NMR were used as input variables.” E.g. caption on Figure 1 (and below in the text).
Author Response
Thank you for the suggestion. The manuscript went an extensive language correction by a professional English translator.
Reviewer 2 Report
The authors have answered quite satisfactory my comments and remarks, but they still need to solve two minor points before publication in Metabolites.
- Previous comment: If a fine multiplicity (d, t, q, dd) is identified in Table S1, I do not see why a J-coupling should not be measured. The authors should complete Table S1.
Authors’ reply: We completed the Table S1 with all values that could be responsibly read from the 1D spectrum. J-values especially for some multiplets were read out from J-res spectra.
I may have been unclear in my previous comment:
- Whenever a fine multiplicity is identified (i.e. the signal is defined as d, t, q, dd), then it is necessarily possible to measure a J coupling (regardless of the pulse sequence, the J is the same, it is only the resolution and signal overlap that change). This means that whenever there is “d” or “t” there MUST be the corresponding value. This has been done indeed for almost all cases, but there are still some errors. For instance in phenylalanine, the signal at 7.44 ppm is a triplet, but authors report two different J couplings, which cannot be.
- Whenever a fine multiplicity cannot be identified (i.e. the signal is defined as m), then it is NOT possible to measure a J coupling. This means that all J couplings given for multiplets (e.g. for 3-hydroxybutyrate or glutamine or phenylalanine) are nonsense.
- Previous comment The resolution of Figures 1 and 2 is quite poor and text is unreadable
Authors’ reply: Thank you for the observation. We replaced the figures and tried to approve the text to be more understable.
Authors just increased the size of the plots, but text is still unreadable (e.g. legend and axis titles are too small).
Reviewer 3 Report
By using the vip-score, at least 3 new biomarkers have been found, for example: lipoproteins, glucose levels and alanine.
On the other hand, after 42 days of hospitalization, the interesting biomarkers are: lipoproteins, glutamine and alanine.
In the Hit map, they found that of great importance as biomarkers are: valine, isoleucine and leucine. These are branched chain amino acids that may be giving us information about the Krebs cycle and the expression of some proteins.
It is very likely that these essential amino acids are not being degraded, therefore it is indicating a problem, in the same way they could be biomarkers. Explain a little this part of biochemistry.
I suggest highlighting this information in the results and discussion.
